# Gaze-behaviors of runners in a natural, urban running environment

**Mark M. Cullen**[1]*, **Daniel Schmitt**[2], **Michael C. Granatosky**[3], **Christine E. Wall**[2], **Michael Platt**[4], **Roxanne Larsen**[1,5]

1 Duke University School of Medicine, Durham, NC, United States of America, 2 Department of Evolutionary Anthropology, Duke University, Durham, NC, United States of America, 3 Department of Anatomy, New York Institute of Technology, Old Westbury, NY, United States of America, 4 Departments of Neuroscience, Psychology, and Marketing, University of Pennsylvania, Philadelphia, PA, United States of America, 5 Department of Veterinary and Biomedical Sciences, College of Veterinary Medicine, University of Minnesota, Saint Paul, MN, United States of America

* mark.cullen@duke.edu

**Data Availability Statement:** All relevant data are within the manuscript and its Supporting Information files.

**Funding:** MMC received the Undergraduate Research Grant from Duke University. Funding for

## Abstract

Gaze-tracking techniques have advanced our understanding of visual attention and decision making during walking and athletic events, but little is known about how vision influences behavior during running over common, natural obstacles. This study tested hypotheses about whether runners regularly collect visual information and pre-plan obstacle clearance (feedforward control), make improvisational adjustments (online control), or some combination of both. In this study, the gaze profiles of 5 male and 5 female runners, fitted with a telemetric gaze-tracking device, were used to identify the frequency of fixations on an obstacle during a run. Overall, participants fixated on the obstacle 2.4 times during the run, with the last fixation occurring on average between 40% and 80% of the run, suggesting runners potentially shifted from a feedforward planning strategy to an online control strategy during the late portions of the running trial. A negative association was observed between runner velocity and average number of fixations. Consistent with previous studies on visual strategies used during walking, our results indicate that visual attentiveness is part of an important feedforward strategy for runners allowing them to safely approach an obstacle. Thus, visual obstacle attention is a key factor in the navigation of complex, natural landscapes while running.

## Introduction

Vision is intricately linked with movement and plays an essential role in guiding locomotion by influencing how an individual reacts to and navigates their environment [1–13]. Previous research suggests that the neuromuscular system adjusts gait parameters based on visual input through two distinct, interrelated strategies: 1) a "feedforward sample controlled method," and 2) an "online control method," [1,2,4–6, 9–11, 13–17].

The feedforward sample controlled method predominates when objects in the environment are stationary, which allows for the identification of obstacles on a path and provides time for

the gaze-tracker was provided through the Duke University Bass Connections program. The funders had no role in study design, data collection and analysis, decision to publish, or preparation of the manuscript.

**Competing interests:** The authors have declared that no competing interests exist.

planning and adjusting gait in anticipation of a stimulus [1, 2, 5, 10, 13, 17]. This allows individuals to utilize their prior experience with the obstacle, or obstacles like it, to safely navigate an approach [1,2,5,10,16,18]. The online control method relies more on peripheral, lower-order processing of visual information and is not based on pre-planning [1, 5, 6, 9, 11]. Hence, individuals can make quick, small adjustments in response to rapidly changing stimuli in the environment [1, 6, 9, 11,19]. Although often discussed as separate processes, these two methods are usually utilized in parallel, facilitating safe navigation of obstacles in one's path. Thus, these two methods allow individuals to regulate step placement on a step-by-step basis and allow them to anticipate and plan a safe path while navigating complex environments [1, 2, 4–6, 9–11,13,17,20].

Considerable empirical research has explored the importance of vision and obstacle avoidance during walking and other athletic events [2, 9,19–33], but there has been less research dedicated to vision in running. Bradshaw and Sparrow (2001) evaluated gaze during running, using head angle rather than a gaze tracker, where they identified similar visual strategies as seen in walking. Their study described three phases of a run associated with visual input and control: (1) an "acceleration from rest" at the beginning, (2) a "global visual control phase" which is a period of adjustment to gain one's heading, and (3) a "local visual control phase" that is established once one is at a comfortable speed and direction of travel and allows for fine adjustments as one moves toward a goal or obstacle. Thus, runners start by accelerating, where they use their global visual control to get a broad sense of their surroundings and environment allowing planning for movement patterns, and then use local visual control to focus on objects of interest in their running path and make small adjustments in order to safely and efficiently navigate different obstacles [23]. The global visual control phase and local visual control phase are comparable to the feedforward sample control method and the online control method respectively and suggest that runners, like walkers, utilize different strategies throughout the action of navigating and clearing an obstacle.

Looking more specifically at the approach and clearance of obstacles during walking, Patla and Greig (2006) suggested that, when faced with an obstacle, focused visual input is essential to make adjustments during the approach rather than during the act of clearing the obstacle itself. This pattern has been observed both in the laboratory and in complex, outdoor environments [1, 2, 5, 9,10,13,17,23, 26, 31]. Regardless of the setting, these studies suggest that individuals most effectively traverse an obstacle when they intermittently sample the object of interest multiple times throughout their approach and have increased, more directed, visual attention when they are approximately two step lengths from the obstacle; supporting a feedforward model during one's approach [2,9,28,29,30]. This demonstrates that, although continuous visual input is important, increased, directed visual attention just prior to approaching an obstacle may also be critical for successful clearance [2, 28, 29, 30, 31].

In the study of peripheral vision, in the context of approaching an obstacle, it plays a role in determining where directed visual attention will be placed. Once this occurs, it appears as is supported above, that directed visual cues from the central visual field predominate [34, 35]. Where peripheral vision appears to dominate is when individuals are clearing an obstacle and transition to "improvisational" online visual control [9,19, 24, 25, 27, 36]. Marigold et al. (2007) observed that when obstacles suddenly appeared within two steps of a person, peripheral vision and an occasional saccade (i.e., a very rapid eye movement directed towards an object) were sufficient to navigate it without the need for a longer span of directed attention on the obstacle. The idea of "attention" was explored further by Weerdesteyn and colleagues (2003, 2005) where they observed higher rates of failure among individuals navigating obstacles while distracted (i.e., multitasking) and in older versus younger individuals when navigating unexpected obstacles.

Additionally, using visual attention to clear obstacles during locomotion becomes more challenging as speed increases. It is thought that as speed increases runners have reduced processing time during their approach, resulting in decreased fluency, stability, and accuracy as they approach an obstacle [22, 37]. Bradshaw and Sparrow (2001) tested the effect that velocity had on visual processing and anticipation by measuring the gait of individuals as they walked, ran, and sprinted toward different obstacles. Their study found that the onset of visual control is directly related to the speed individuals' run toward an obstacle, demonstrating that sprinting participants had a later onset of visual control. These results coupled with a later experiment on long jumping [26] suggest that when individuals move at higher rates of speed, there may be less visual control and a reduced ability to anticipate hazards in their path.

Visual attention can be defined by determining visual fixation on an object. A fixation is generally taken to represent the intentionality in gaze and is a specific period of time in which an individual obtains relevant information about their environment [2, 38, 39, 40]. Although the time that defines a single fixation remains debated in the literature, with numbers ranging from 80–150 ms, a number of studies using similar methods to ours use a value of 99 ms as representative of intentionality in gaze behavior [2, 38, 39, 40]. Thus, in the current study, we also use 99 ms to quantify the timeframe of a fixation, which allowed us to explore the role of vision during the approach of an obstacle (i.e., a sidewalk curb) while running in an outdoor environment.

Specifically, this study attempts to answer the question, "Do runners fixate on a sidewalk curb when running in an urban setting?" To address this question and to focus our investigation, this study utilized a gaze-tracker and a custom-made algorithm that allows for comparisons between computer and human analyses of data to explore two primary hypotheses (our null hypotheses) and their alternates.

### Hypothesis 1

$H_{1o}$.   The majority of runners in our study will use an online sampling method as evidenced by the absence of fixations on the obstacle throughout the entirety of the run.

$H_{1a}$.   The majority of runners in our study will use a feedforward sampling approach as evidenced by fixations on the obstacle throughout the approach phase of their run as they move toward the obstacle (i.e., the sidewalk curb).

$H_{1b}$.   The majority of runners in our study will utilize a combination of feedforward and online sampling strategies as indicated by the occurrence of interspersed fixations on the obstacle with times of no fixation on the obstacle.

### Hypothesis 2

$H_{2o}$.   The average velocity of the runner has no impact on the number of fixations on the obstacle along the run.

$H_{2a}$.   The faster a runner travels the fewer fixations on the obstacle throughout a run, and thus the slower a runner travels the more fixations on the obstacle throughout the run.

## Methods

### Experiment, participants, and equipment

Ten runners (5 males and 5 females; Leg Length 95.19 +/- 7.37-cm; Table 1) were recorded running on a 20-m long sidewalk path with an obstacle (i.e., a curb that was 0.15-m high), a natural, urban running environment (Fig 1). Participants wore a gaze-tracker, modified with an adjustable visor (Zhi Jin, Hong Kong, CN), and its small backpack (CamelBak, Petaluma,

**Table 1. Demographic characteristics and average running velocity for the subject pool used in this study (n = 10) as an aggregate and divided by gender.**

| Characteristic | Sample (n = 10) | Range | Male (n = 5) | Female (n = 5) |
|---|---|---|---|---|
| Age (yrs.) | 22.50 +/- 2.76 | 21–29 | 22.80 +/- 3.49 | 22.20 +/- 2.17 |
| Height (m) | 1.74 +/- 0.12 | 1.55–1.93 | 1.84 +/- 0.06 | 1.64 +/- 0.08 |
| Weight (kg) | 75.98 +/- 16.97 | 52.16–115.67 | 85.28 +/- 17.54 | 60.33 +/- 11.16 |
| Velocity (m/s) | 3.02 +/- 0.14 | 2.81–3.18 | 3.06 +/- 0.14 | 2.97 +/- 0.15 |
| Experience (yrs.)[1] | 5.60 +/- 2.91 | 1–7 | 5.00 +/- 2.35 | 6.20 +/- 3.56 |

[1]Experience was defined as the year where participants felt they began running subtracted from their age at the time of the trials.

CA, USA) while jogging or warming up in their normal manner (Fig 2). Running with a gaze tracker has the potential to induce movement that can affect crosshair position and cause crosshair drop-outs. To minimize this effect the shield and the head band secured the glasses portion of the unit securely to the runner's face, limiting its motion.

Data were collected as participants ran at a self-selected distance-running pace and took place at dawn and dusk to limit light exposure. At approximately 15-m into the run, participants encountered and cleared the curb. Participants ran up to 15 laps along the sidewalk and in a loop back to the start of the path, with each lap considered a "trial." All protocols were approved by the Duke University IRB (protocol 2017–0947) and the individual in this manuscript (Fig 2) has given written informed consent (as outlined in the PLOS consent form) to publish these case details.

The current study uses methods similar to previous studies including the use of eye tracker equipment [2, 3, 11, 33], and the use of a natural environment to study human locomotion [33]. The gaze-tracker (Omniview-TX Head-Mounted Eye Tracking, 30 Hz, ISCAN Inc., Woburn, MA, USA) was composed of three different cameras. The two nearest the eyes

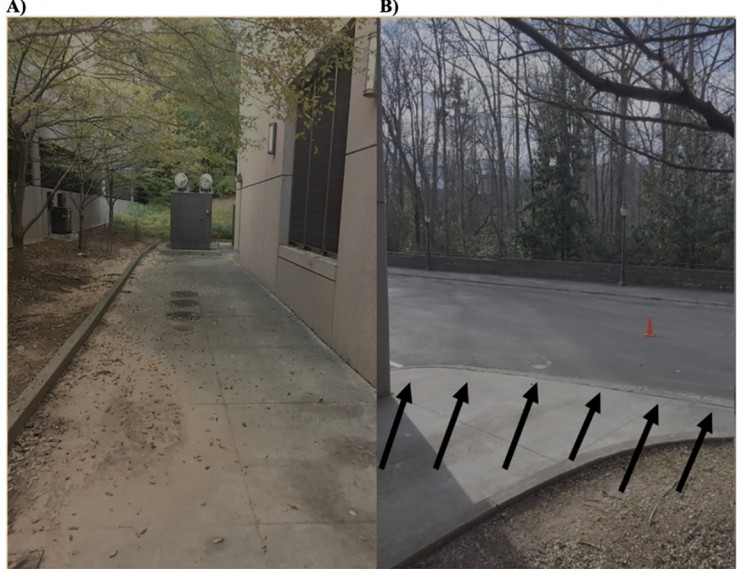

**Fig 1. Natural running environment showing Curb: This is a natural running environment in a typical city in the United States.** The track was made of concrete. A) The track participants ran as they approached the curb. B) Arrows designate the curb participants cleared. The cone in the background designated the turning point for runners to start another lap.

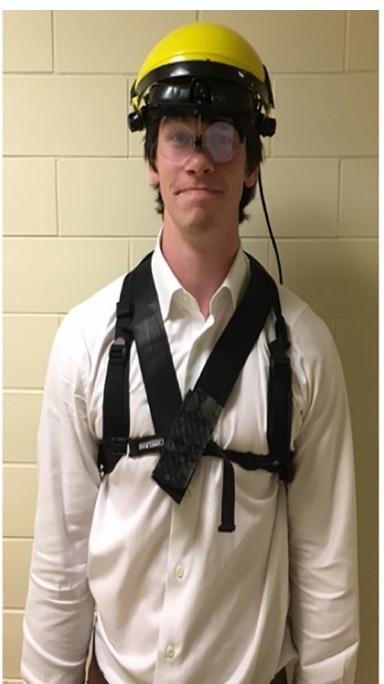
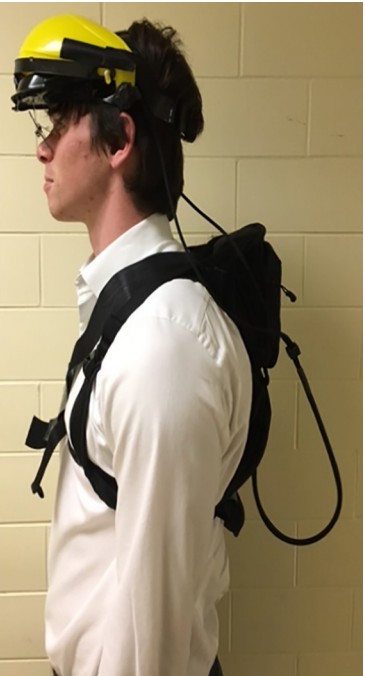

**Fig 2. ISCAN Omniview-TX Head-Mounted Eye tracking device worn by one of our authors (MC).** This has been modified by combining it with a face shield (portion in yellow), allowing the glasses to be held tightly against the face even while moving. In addition, the transmitter was placed into a backpack, which allowed for a natural place for the transmitter to be housed while an individual was running.

measured the right and left eye positions of the runner utilizing infrared light, and the third camera displayed the point of view of the runner as they traveled down a path. The gaze-tracker did not impair the vision of the runner, and the form of the runner was not impacted by the telemetry equipment worn on the participants' backs. To calibrate the gaze-tracker, before running their trials participants looked at an ISCAN calibration board containing five points. At a distance of 3.0-m, participants first looked at the center point, followed by the top right, left and finally bottom left and right points consecutively. A distance calibration was also performed on the center calibration point to ensure calibration at different distances from the board. This resulted in an accurate projection onto the video frames of where the participants were looking. This projection was represented as a crosshair in each video frame (Fig 3).

Inclusion criteria for final analysis was based on several factors. In order to have a balanced sample to capture variation we selected six trials for each subject. We reduced the effect of novelty and learning during each run by choosing trials, across the entire range of the run for each participant (two at the beginning, two in the middle, and two at the end). To ensure quality in our data, the crosshair was recorded in the participants' field of view for at least 84% of a trial. This was meant to reduce crosshair dropouts secondary to micromovements of the headset, interruption in the infrared camera secondary to ambient light exposure, as well as loss due to participant's eyes looking beyond the peripheral limits of the tracking system (i.e., it was outside the view of the two small cameras positioned on the participant's eyes). The majority of trials met this criterion. If one of the six trials selected had a cursor in fewer than 84% of the frames it was not used and another one from the same part of the trial set was selected. Therefore, a total of 60 trials with the cursor present in greater than 84% of the frames were analyzed (six per subject).

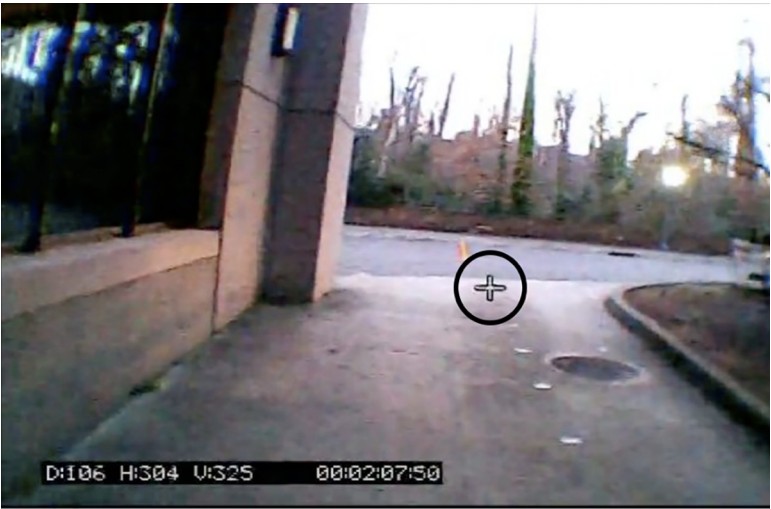

**Fig 3. The crosshair represents where the participants were looking in their visual field during the run.** This was digitized throughout each of the participants trials. The location of the digitized crosshair was used to determine if the participants were looking near or on the curb.

Objects in videos were digitized utilizing DLTv5 [41]. The crosshair (at its center) and curb (the four corners of the curb creating a quadrilateral) were digitized for the entire length of each trial, from the beginning of the running trial, to the time the curb disappeared from the participant's vision at the end of the trial (Fig 3). All trials were examined for fixations. Those trials where the subject never fixated on the curb at any point were noted and included in analysis of the frequency of trials with no fixations (indicating on-line sampling throughout) and included in the analysis of fixation counts in bins, but not the timing of last fixation.

## Algorithm

In studies of gaze it is a challenge to measure a fixation in a 2D projection of a visual field, i.e., to determine how close to the object the cursor must be to be scored as looking at the object. To resolve this issue, we designed an algorithmic automated system that incorporates system-based error and human-based (45 raters) judgments of what counted as attention to the curb. This was a multi-step process. First, to create a reasonable estimate of the area in the frame that would represent if the runner had placed their vision on the curb, the original digitized space was expanded slightly to include the error associated with the gaze tracker and potential calibration error (resulting in a quadrilateral that was slightly larger than the size of the curb itself). The amount of calibration error reported in the literature for the system used in this experiment and similar gaze tracking systems varies depending on the model and context in which it was used [11,27,33,42,43] (0.50˚-2.00˚). To determine an appropriate calibration error to be added around the digitized curb, an experiment was completed where the digitized curb was expanded on all sides at increments of 0.25˚ in error magnitude starting at 0.50˚ and moving to 2.00˚ (Fig 4).

To better quantify and identify the different regions near the curb, the video frame encompassing the field of view of the participant, was converted from degrees (76˚ x 52˚, length by width) to pixels utilizing the conversion factor of 1˚: 8.62 pixels in the X-direction and 1˚: 9.23 pixels in the Y-direction (equating to a total of 655 pixels x 480 pixels). If the crosshair fell within one of these incremental zones (curb plus pixel, error of magnitude), then the algorithm

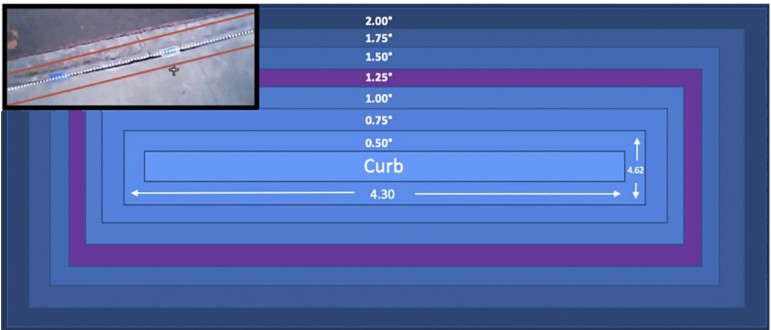

**Fig 4. The digitized curb created from the algorithm based on magnitudes of error.** The inset shows an actual video frame with the digitized curb (white dotted line, which equals 0.00˚ (the curb), and the red solid line equal to 2.00˚ magnitude of error, which was the maximum magnitude of error of the gaze-tracker found in similar studies. These error magnitudes were used to make algorithms, which were then compared to human raters to determine which error magnitude most closely replicated the interpretations of human raters. To create each increase in error for the curb, the error magnitudes (commonly expressed in degrees), were converted to pixels, allowing error to be matched with the units used in DLTdv5. The conversion factor for this chart in the X direction was 1 degree: 8.62 pixels and in the Y direction was 1 degree: 9.23 pixels. The central rectangle represents the curb (0.00˚). For each successive increase in error magnitude from the original digitized curb through the largest error magnitude of 2.00˚ we added 4.30 pixels in the X direction and 4.62 pixels in the Y direction. The 1.25˚ error magnitude, shown in purple, was found to be the most representative of the interpretations of the human raters.

indicated the runner looked at the curb (a "hit") and if the crosshair was absent then the algorithm indicated the runner was not looking at the curb (a "miss"). This created an absence/presence tally across the running trials.

Then, to further automate the algorithm and evaluate it against human raters, 45 human raters examined twenty frames that had been evaluated by each algorithm (accounting for varying amounts of error) and rated each frame as a "hit" and a "miss." Variability was noted in some of the frames (some raters determined that a frame was a "hit" while others determined it was a "miss") and so a 65% threshold was determined to distinguish a hit from a miss. The threshold was calculated by using a Z-test and setting a 95% confidence interval on the hit/miss results from the group of raters (n = 45). Although more conservative than other choices, our algorithm determined "hits" that we were certain our raters felt represented the runner was looking at the curb. This allowed for comparison between the human rater's judgment, often the standard in studies like this, and the designations given by the incremental zones of error magnitude within the algorithm. This two-part approach allowed for an algorithm to be chosen that most closely represented the perception of these human raters. During this comparison, when the raters called the frame a "hit" but the algorithm indicated it as a "miss," this was defined as a "false miss (FM)." And when the raters called a frame a "miss" but the algorithm indicated it as a "hit" this was called a "false hit (FH)."

The third step in this process was to evaluate the concordance of each algorithm with human judgment. When an algorithm produced a large number of FMs, it was considered too conservative and when an algorithm produced a large number of FHs, it was considered too liberal. To measure reliability between raters' decisions and computer decisions a Cohen's Kappa (CK) [44] and a Goodman and Kruskal's lambda [45] were both utilized. Both methods measure the agreement between each set of data and accounts for any agreement that might arise due to chance. A value for either statistic closer to 1.0, signifies higher levels of agreement between the raters and each algorithm. Importantly, a value of 0.0, indicates the two data sets are independent from one another. The difference between the two is that the Goodman and Kruskal's lambda is somewhat more conservative than Cohen's Kappa. With a Cohen's Kappa of 0.89 and a Goodman and Kruskal's lambda of 0.85, it was determined 1.25˚ of calibration

**Table 2. Summary of each Algorithm's "Hit/Miss" Interpretations and Cohen's Kappa measuring the reliability between the Algorithm and the Rater.**

| Hit ≥% | Calibration Error (°) | 0.50 | 0.75 | 1.00 | 1.25 | 1.50 | 1.75 | 2.00 |
|---|---|---|---|---|---|---|---|---|
| 65 | False Miss | 3 | 1 | 0 | 0 | 0 | 0 | 0 |
| | False Hit | 0 | 0 | 0 | 0 | 2 | 2 | 2 |
| | Cohen's Kappa | 0.53 | 0.78 | **0.89** | **0.89** | 0.70 | 0.53 | 0.53 |

error magnitude produced results most similar to the human raters. This error magnitude was then used on the remainder of our participants to determine if the participant was looking at the curb (Table 2; Fig 3). The error magnitude calculated for our study is similar to the magnitude of error found in Matthis and colleagues (2018), which is the most similar to our experimental design in that it was studying human locomotion in a natural environment utilizing a gaze-tracker.

## Data analysis

This definition of a fixation follows previous studies [2, 38,39,40], and in our study was determined as the point at which the runner's gaze fell within the curb area (with 1.25° of calibration error applied), for at least three consecutive frames (defined as 99 ms). Therefore, a fixation in this study is taken to represent intentionality in the gaze of the runner and thus exhibited a period in which the runner was obtaining relevant information about their environment.

Sex-specific differences in the number of fixations per trial were assessed with a Mann-Whitney-U test. To better investigate fixations across the length of the running trials, each participant's trials were divided into fifths based on the total time of the trials (each time increment was identified as a "Bin" yielding Bins 1, 2, 3, 4, and 5), which allowed for a summed average of fixations between individuals at different portions of their run. Bin 1 and 2 represent the early time increment, Bin 3 represents the middle time increment, and Bin 4 and 5 represent the end time increment (which ended when the curb was no longer in the field of view of the gaze tracker). Goodness of fit tests were utilized to compare the summed counts of fixations in each bin.

A series of Spearman's rank correlation analyses were used to assess the relationship between runner speed and total number of fixations on the obstacle, and the relationship between runner speed and the timing of the last fixation. Timing was measured as a percent of the trial in which the last fixation occurred (meaning that for each trial with a fixation, the time period the last fixation occurred was normalized to the total time of the run and compared with all of the other trials). All statistical tests were conducted in PRISM (Graphpad Software, San Diego, CA).

## Results

### Total fixations, fixations across each trial, and timing of the participant's last fixation

Fixations were observed in 80% (n = 48 out of 60) of trials. No subject had a complete set of trials with no fixation; that is all subjects fixated on the curb during some of their trials. Trials without fixations were fairly evenly distributed across subjects with half of the subjects fixating in every trial, one subject with one trial with no fixations, one subject with two trials with no fixations, and three subjects with three trials with no fixations. Thus, a total of 12 trials out of 60 appeared to use exclusively online sampling.

On average, participants fixated on the curb 2.4 ± 1.9 times during a trial (Fig 4). The number of fixations during a trial did not vary significantly (Mann-Whitney U = 11, p = 0.79) between males (2.4 ± 1.9) and females (2.3 ± 1.3). No significant difference was found when the summed counts were compared across bins using a Chi-Square Goodness of Fit (χ = 4.818; p = 0.31; Fig 5).

### Timing of last fixation for each trial

On average, the timing of the last fixation occurred at 59.2% ± 24.0% of the total trial time (Fig 4). No significant differences (Mann-Whitney U = 11, p = 0.84) were observed between the percent of trial in which the fixation last occurred for males and females (males = 62.9 ± 18.1% versus females = 55.5 ± 30.6%; $P$ = 0.8413). However, the 60% average is driven in part by a bimodal distribution with half of the subjects showing the average last fixation occurring at approximately 40% of the run (roughly six meters from the curb) and the other half of the subjects average last fixation occurring at approximately 80% of the run (roughly 3 meters from the curb). It should be noted that one subject fixated intensely only at the end of the run in all their trials and may have driven the high values for the 80% average group.

### Correlation analyses

Results of the Spearman's rank correlation estimates demonstrated that there is no significant relationship between running speed and the timing of the last fixation (R = -0.02, $P$ = 0.97). There was, however, a significant negative relationship between running speed and the average number of fixations within a trial (*r = -0.76; P* = 0.01; Fig 6).

### Discussion

This study developed and tested hypotheses, based on previous laboratory and athletic field studies, about how human runners interact visually with a common obstacle (a typical sidewalk curb). The goal was to explore how much visual attention runners gave to a relatively common and easy to navigate obstacle, as opposed to a sudden or complex obstacle. To do this we recorded and quantified the fixations of runner's as they approached the curb and

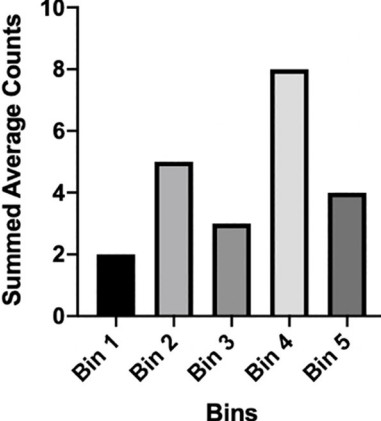

**Fig 5. Summed average counts of fixations across participants split into bins.** No significant difference from the expected frequency noted in each bin utilizing a chi-square goodness of fit test.

## Correlation of Speed to Average Number of fixations

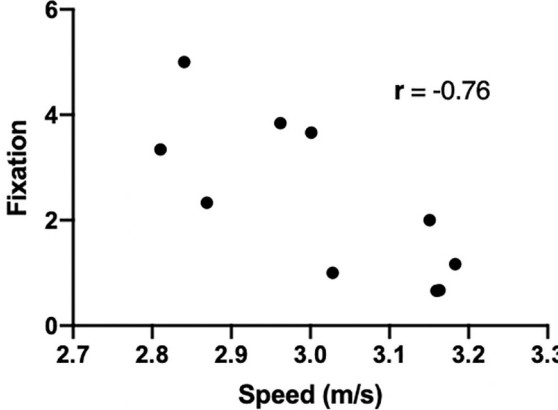

**Fig 6. The relationship between the average number of fixations for each participant and their average speed.**
There is a strong negative relationship between how fast a participant was running and the number of average fixations
($r = -0.76$, $P = 0.01$).

identified where these fixations took place along the length of the run. Out of the 60 trials
observed from our participants, 80% of the trials had at least one fixation on the curb, every
participant had a trial with at least one fixation, and, on average, runners fixated on the cub
2.4 ± 1.5 times during each run. This rejects hypothesis $H_{1o}$ that subjects would not fixate on
the curb at all. Online-sampling (as evidenced by the absence of fixation) is not a primary
mechanism of attention at least during the majority of the duration of those runs.

Subjects had fixations throughout most of the trial and there were no differences in average
fixation numbers between female and male participants. These results suggest that fixating on
a region of interest in one's path or environment at least once or twice during the approach
phase is a common strategy among our participants and is important for implementing clear-
ance maneuvers, proactive adjustments of body position, and route planning during running.
Thus, the feedforward method of attention is seen here as a central part of obstacle manage-
ment during running in a natural urban setting.

The results presented herein suggest two things about the latter portion of a run. First, on
average the highest number of summed counts of fixations occurs between 60% and 80% of
the run (Fig 4), a pattern consistent with previous studies of walking subjects [1, 2, 5, 9,10,
13,17, 23, 26, 31]. Second, this study found that individuals' average last fixation on the curb
occurred in the last 20% - 40% of their run. There was no correlation between the runners'
final fixation time and their running speed, indicating that regardless of how fast a runner is
moving, they still fixate near the middle of their run to judge the distance to an obstacle, and
then proceed to clear an obstacle without the need for further fixations. The timing of one's
final fixation is also consistent with the feedforward model suggested by previous studies in
walkers [1, 2, 5,10, 13,17], as it suggests that once runners are over halfway through their run,
they have already pre-planned their path, adjusted their gait for the oncoming obstacle, and
are looking onward for the next obstacle in their path. Further, it could be suggested that at
this time in the run it may be advantageous for runners to transition from the more "proac-
tive" feedforward aspect of vision to the more "improvisational" online control method, allow-
ing runners to make quick on-the-fly adjustments to their gait to ensure safe navigation of the
obstacle, while also planning for the next oncoming obstacle [6, 9, 11].

A negative correlation was found between the running speed of an individual and the total number of fixations within a trial. This suggests that faster runners who approach the curb relatively quickly have less time for feedforward adjustments and rely more heavily on the online control method in order to "fine-tune" their movements as they get closer to the obstacle [6, 9, 11].

There are several limitations to the current study. The relatively small sample size of 10 participants, all of whom were young and associated with Duke University may not be representative of the general running population. Including older runners, as well as those with different levels of running experience would be an important area for further research. However, these comparisons are beyond the scope of this study. In addition, when possible, we encourage future researchers to use more than 20 different gaze images to develop the criteria for their algorithm, as this sample size could have caused bias in our kappa and lambda statistics. It is also the case that our gaze-tracker utilizes infrared reflection lenses and that the use of this system during a running task is prone to movement of these lenses and thus could have disrupted our previous calibration.

To further understand the role of vision during obstacle clearance it would be worth examining saccades (i.e., shorter bouts of visual attention) as individuals cleared the obstacle [11]. In addition, studying the role of the runner's peripheral vision in future studies is warranted to better characterize the online control method and the feedforward method which appear to predominate during clearance and approach, respectively. It would also be valuable to measure the step length of runners throughout the trial [46] and further, to examine if those that fixate on the curb more at the end of the trial change step length similarly to those that do not look at the curb as frequently. Additionally, using obstacles of different height similar to previous studies in walkers [26, 36] would be an interesting addition to a future study. Recently, Lucas-Cuevas et al. [47] demonstrated that, during treadmill running, the further that one's gaze is from a region of interest, the worse one's gait mechanics became while running and the less comfort runners had during that part of the experiment. It would be productive to look into the distance runners tend to focus outside of a ROI during safe obstacle navigation to get a sense as to how far from an obstacle individuals can focus and still obtain enough information to effectively and safely move down a path with a perturbation.

This study supports the hypothesis that runners will use a combination of feedforward and online sampling methods to navigate obstacles in a natural urban setting. For subjects in this study, visual attentiveness is part of a feedforward strategy for runners to plan and safely navigate an obstacle during the first 40% - 80% of the run, after which the runner shifts to an online-sampling method in which the curb is not a primary visual focus. Overall, this study adds to our understanding of the role of visual attention in human runners and is essential for understanding broader aspects of locomotor control and decision-making for runners as they navigate outdoor challenges.

## Supporting information

**S1 Data.**
(XLSX)

**S2 Data.**
(XLSX)

## Acknowledgments

The authors thank Angel Zeininger, Aidan Fitzsimons, and Megan Snyder for their contributions to the conception and design of this work.

## Author Contributions

**Conceptualization:** Mark M. Cullen, Daniel Schmitt, Michael C. Granatosky, Michael Platt, Roxanne Larsen.

**Data curation:** Mark M. Cullen, Daniel Schmitt, Christine E. Wall, Roxanne Larsen.

**Formal analysis:** Mark M. Cullen, Daniel Schmitt, Michael C. Granatosky, Christine E. Wall, Roxanne Larsen.

**Funding acquisition:** Daniel Schmitt, Michael C. Granatosky, Michael Platt, Roxanne Larsen.

**Investigation:** Mark M. Cullen, Daniel Schmitt, Michael C. Granatosky, Roxanne Larsen.

**Methodology:** Mark M. Cullen, Daniel Schmitt, Michael C. Granatosky, Christine E. Wall, Roxanne Larsen.

**Project administration:** Daniel Schmitt, Michael Platt, Roxanne Larsen.

**Resources:** Daniel Schmitt.

**Supervision:** Daniel Schmitt, Michael C. Granatosky, Christine E. Wall, Michael Platt, Roxanne Larsen.

**Validation:** Mark M. Cullen, Daniel Schmitt, Christine E. Wall.

**Visualization:** Mark M. Cullen, Daniel Schmitt, Christine E. Wall, Roxanne Larsen.

**Writing – original draft:** Mark M. Cullen, Daniel Schmitt, Roxanne Larsen.

**Writing – review & editing:** Mark M. Cullen, Daniel Schmitt, Michael C. Granatosky, Christine E. Wall, Michael Platt, Roxanne Larsen.

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
