## [Decision Letter · Decision Letter 0]

16 Apr 2020

PONE-D-20-07008

Gaze-Behaviors of Runners in a Natural, Urban Running Environment

PLOS ONE

Dear Mr. Cullen,

Thank you for submitting your manuscript to PLOS ONE. After careful consideration, we feel that it has merit but does not fully meet PLOS ONE’s publication criteria as it currently stands. Therefore, we invite you to submit a revised version of the manuscript that addresses the points raised during the review process.

We would appreciate receiving your revised manuscript by May 31 2020 11:59PM. To enhance the reproducibility of your results, we recommend that if applicable you deposit your laboratory protocols in protocols.io, where a protocol can be assigned its own identifier (DOI) such that it can be cited independently in the future. For instructions see: http://journals.plos.org/plosone/s/submission-guidelines#loc-laboratory-protocols

We look forward to receiving your revised manuscript.

Kind regards,

Nizam Uddin Ahamed, PhD

Academic Editor

PLOS ONE

Journal Requirements:

1. Please amend your manuscript to include your abstract after the title page.

Reviewers' comments:

Reviewer's Responses to Questions

**Comments to the Author**

1. Is the manuscript technically sound, and do the data support the conclusions?

Reviewer #1: Partly

Reviewer #2: Yes

2. Has the statistical analysis been performed appropriately and rigorously? 

Reviewer #1: Yes

Reviewer #2: Yes

3. Have the authors made all data underlying the findings in their manuscript fully available?

Reviewer #1: No

Reviewer #2: Yes

4. Is the manuscript presented in an intelligible fashion and written in standard English?

Reviewer #1: Yes

Reviewer #2: Yes

5. Review Comments to the Author

Reviewer #1: The study attempts to answer the question of whether runners fixate on obstacles such as sidewalk curbs when running in an urban road setting. To answer the question, authors followed a dual-process visual control framework, used a binocular gaze-tracker, and developed a customized algorithm to test gazing behaviors among runners in two primary hypotheses. Results showed that runners tended to use a combination of online and feedforward sampling control strategies and that faster average running pace led to fewer fixations on the obstacle of sidewalk curb. In many ways the manuscript is interesting, particularly regarding the development of customized eye-tracking algorithm in dealing with visual fixations on objects in 3D space. It is also well written. With that said, I do have several concerns that could help enhance the manuscript. The concerns were elaborated below.

First, I have concerns about employing the current eye-tracking system in a running task. Based on what I can find given the information authors provided, the eye-tracking system seems to involve infrared (IR) reflection lenses (i.e., the hot mirrors) in tracking the eyes. Such a design of IR lenses, compared to those embedded in the glasses frames in some other eye-tracking brands, may be less fit to studying the running task. It is because running movements involve constant bodily acceleration that can dislocate previously calibrated IR reflection lenses, which is usually connected to the glasses frame via weak plastic connectors. I understand that, if I am correct about the hardware, the authors may have limited power in attenuating the above issue associated with the equipment design, but I encourage them to fully acknowledge such difficulties and include a picture of the exact eye-tracking system used.

Another challenge of tracking runners’ vision comes from the fact that runners generally need to keep their upper bodies (including heads) relatively stable during running to maximize movement efficiency. That is, keeping the head still during running requires more visual search to be achieved by replacing head movements with eyeball movements, which makes the pupils more likely to appear at off-center locations in the eyes. For the majority of eye-tracking systems utilizing the ‘pupil to CR (corneal reflection)’ technique, big eyeball movements would cause technical issues by making the IR dots more likely to fall off the pupil region and by reshaping the appearance of the pupil from a circle to a oval shape, which is harder for the algorithm to recognize. That is to say, the task would cause increased likelihood for tracking loss. Although authors reported the inclusion criterion of 84% (l.223), I worry about potential dependence between tracking loss and gazing behaviors in the running task, which can result in bias regarding evidence analyzed and conclusions. I would encourage authors to do some analyses comparing those low-tracking rate trials (e.g., the lowest 6 trials) with the current evidence analyzed so that we can either eliminate or confirm such a possible bias, either way it would be a contribution.

Furthermore, although the authors did a good job reviewing literature of foveal vision in relevant task models (to some extent, I feel that the authors overly cited the literature in several locations in the Introduction, so I suggest to cut off some redundant ones), the literature on peripheral vision is largely missing. It makes sense that the foveal vision should be a focus of literature review given the use of eye-tracking, which is really foveal vision tracking. However, visual attention can come from or even be dominated by peripheral vision. It is likely that in some instances runners rely more on peripheral vision in monitoring environments. I would recommend authors give a light weighted review (and also discussion) on peripheral vision. A recent review from Vater et al. would be a good start place.

Vater, C., Williams, A. M., & Hossner, E. J. (2019). What do we see out of the corner of our eye? The role of visual pivots and gaze anchors in sport. International Review of Sport and Exercise Psychology, 1-23.

Since the algorithm is supposed to be a major contribution of the manuscript, I have some suggestions to enhance it: 1. try to include Goodman and Kruskal’s Lambda in addition to Cohen’s Kappa. Lambda could give another view of the ‘rater agreement’; 2. explicitly acknowledge the small number of frames (i.e., 20, l. 279) used in testing the algorithm and encourage future studies to increase it. Such a small number may generate biased Kappa and Lambda estimates and is subjected to selection risks (i.e., why this 20 out of potentially hundreds of frames? why not randomly draw from the pool of available frames?).

Lastly, I have some comments on the results section. Authors non-parametrically tested a small sample (n=10). None of the results in Section 3.1 and 3.2 reached significance (p < .05), which is not surprising given the compromised statistical power from both directions. The only significant results came from a parametric test (e.g., simple regression) in Section 3.3. To be consistent in authors’ testing approach, shouldn’t they replace the a simple regression (which basically gave estimates on Pearson r) with Spearman Rank correlation estimates?

Reviewer #2: The study tackled an interesting topic on gaze behavior of the runners in facing the natural obstacles in terms of pre-planning vs. online planning which is important to the respective field, the statistical analyses and results were sound and the manuscript was well-written.

6. PLOS authors have the option to publish the peer review history of their article (what does this mean?). If published, this will include your full peer review and any attached files.

Reviewer #2: No

---

## [Author Response · Author response to Decision Letter 0]

24 Apr 2020

We want to thank the editor and the two anonymous reviewers for comments and suggestions. We were able to accommodate every suggestion and we think that the changes we made have improved the manuscript. We are grateful for the opportunity to submit a revised version and hope that you find this manuscript acceptable for publication.

Reviewer #1: The study attempts to answer the question of whether runners fixate on obstacles such as sidewalk curbs when running in an urban road setting. To answer the question, authors followed a dual-process visual control framework, used a binocular gaze-tracker, and developed a customized algorithm to test gazing behaviors among runners in two primary hypotheses. Results showed that runners tended to use a combination of online and feedforward sampling control strategies and that faster average running pace led to fewer fixations on the obstacle of sidewalk curb. In many ways the manuscript is interesting, particularly regarding the development of customized eye-tracking algorithm in dealing with visual fixations on objects in 3D space. It is also well written. With that said, I do have several concerns that could help enhance the manuscript. The concerns were elaborated below.

We want to thank the reviewer for their compliments and the suggestions for ways to improve the paper. We have been able to take all these suggestions into account and believe that the paper is significantly improved.

First, I have concerns about employing the current eye-tracking system in a running task. Based on what I can find given the information authors provided, the eye-tracking system seems to involve infrared (IR) reflection lenses (i.e., the hot mirrors) in tracking the eyes. Such a design of IR lenses, compared to those embedded in the glasses frames in some other eye-tracking brands, may be less fit to studying the running task. It is because running movements involve constant bodily acceleration that can dislocate previously calibrated IR reflection lenses, which is usually connected to the glasses frame via weak plastic connectors. I understand that, if I am correct about the hardware, the authors may have limited power in attenuating the above issue associated with the equipment design, but I encourage them to fully acknowledge such difficulties and include a picture of the exact eye-tracking system used.

Thank you for your comment. This was a major concern for us as well. Yet we really wanted to expand the use of these systems into dynamic settings and we hope we made adjustments that make this system work well. The current system is the most modern designed by iScan and we worked with them to make it as sturdy as possible for this goal when we purchased. In addition, we designed the helmet with the tight fitting band on the head that limited movement and firmly affixed the glasses in places (without discomfort). We have inserted a new figure, Figure 2 in the manuscript, of one of our authors (MC) wearing our gaze-tracker, helmet and backpack. We did all our runs at dusk and dawn to minimize outdoor light exposure. We have expanded on that in the methods (starting on line 194) and have acknowledged these limitations in our methods and discussion (line 458-461). In addition, recognizing the limits we considered larger regions of gaze (the curb broadly speaking and with error incorporated) to avoid some of those issues. We hope the reader will consider these limitations and that future studies will further address design. 

Another challenge of tracking runners’ vision comes from the fact that runners generally need to keep their upper bodies (including heads) relatively stable during running to maximize movement efficiency. That is, keeping the head still during running requires more visual search to be achieved by replacing head movements with eyeball movements, which makes the pupils more likely to appear at off-center locations in the eyes. For the majority of eye-tracking systems utilizing the ‘pupil to CR (corneal reflection)’ technique, big eyeball movements would cause technical issues by making the IR dots more likely to fall off the pupil region and by reshaping the appearance of the pupil from a circle to a oval shape, which is harder for the algorithm to recognize. That is to say, the task would cause increased likelihood for tracking loss. Although authors reported the inclusion criterion of 84% (l.223), I worry about potential dependence between tracking loss and gazing behaviors in the running task, which can result in bias regarding evidence analyzed and conclusions. I would encourage authors to do some analyses comparing those low-tracking rate trials (e.g., the lowest 6 trials) with the current evidence analyzed so that we can either eliminate or confirm such a possible bias, either way it would be a contribution.

Thank you for your comment and we feel that this is a great point. We went through all of our trials that were excluded either because they were not at the correct portion of the run (beginning, middle, or end) or did not meet our 84% inclusion criteria. We found that 49% of our rejected trials met the 84% or greater inclusion criteria. As a result both low and high cursor trials were excluded. When we expanded this to 80%, it included 71% of our sample, and finally when expanded to 75%, this included 96% of our sample. We hope that this data shows that the trials that were excluded had the crosshair present for a similar amount of time as the trials we included and that often it was the time of the run (beginning, middle, and end) that determined what trial was chosen rather than a trial having the highest percentage of crosshair presence. The statement “For those with more than six, the trials with the six highest percentages of crosshair presence were included,” has been removed and the methods (lines 236- 247) have been modified to reflect that fact that trials were chosen from the beginning, middle, and end of the run, and that each of these met our 84% inclusion criteria. We would like to emphasize that this does not mean that the trials chosen for a particular individual included the highest percentage of crosshair presence, it was the fact that it met our criteria of having the crosshair for greater than 84% of the trial and the trial was at the correct portion of the run. 

Furthermore, although the authors did a good job reviewing literature of foveal vision in relevant task models (to some extent, I feel that the authors overly cited the literature in several locations in the Introduction, so I suggest to cut off some redundant ones), the literature on peripheral vision is largely missing. It makes sense that the foveal vision should be a focus of literature review given the use of eye-tracking, which is really foveal vision tracking. However, visual attention can come from or even be dominated by peripheral vision. It is likely that in some instances runners rely more on peripheral vision in monitoring environments. I would recommend authors give a light weighted review (and also discussion) on peripheral vision. A recent review from Vater et al. would be a good start place.

Vater, C., Williams, A. M., & Hossner, E. J. (2019). What do we see out of the corner of our eye? The role of visual pivots and gaze anchors in sport. International Review of Sport and Exercise Psychology, 1-23.

Thank you for the reference this helped us really consider this issue. We have added references to the manuscript and have added to our introduction more on peripheral vision during the approach (lines 118-121) and have emphasized its importance during clearance of the obstacle (lines 121-122). We added a sentence on the need for further exploration of the study of peripheral vision in running to better characterize our understanding of the approach and navigation of obstacles as human’s run toward them in the discussion (line 465-467).

Since the algorithm is supposed to be a major contribution of the manuscript, I have some suggestions to enhance it: 1. try to include Goodman and Kruskal’s Lambda in addition to Cohen’s Kappa. Lambda could give another view of the ‘rater agreement’; 2. explicitly acknowledge the small number of frames (i.e., 20, l. 279) used in testing the algorithm and encourage future studies to increase it. Such a small number may generate biased Kappa and Lambda estimates and is subjected to selection risks (i.e., why this 20 out of potentially hundreds of frames? why not randomly draw from the pool of available frames?).

This is a great suggestion, thank you. We have calculated the Goodman and Kruskal’s lamba and have found that the pattern is fundamentally the same. We have included the value in the methods section with the Cohen’s Kappa (line 322). We agree that future studies should add more frames where possible. We would like to acknowledge that we had a limited number of frames with diversity (i.e. frames were meant to be a representative sample of some where the individual was clearly not or was looking at the curb, as well as a representative sample of some that were more ambiguous). We have acknowledged this limitation at the end of the in the manuscript and have a sentence encouraging others to increase the number of frames in the future (lines 456-458). 

Lastly, I have some comments on the results section. Authors non-parametrically tested a small sample (n=10). None of the results in Section 3.1 and 3.2 reached significance (p < .05), which is not surprising given the compromised statistical power from both directions. The only significant results came from a parametric test (e.g., simple regression) in Section 3.3. To be consistent in authors’ testing approach, shouldn’t they replace the a simple regression (which basically gave estimates on Pearson r) with Spearman Rank correlation estimates?

Thank you for your comment. We have changed our parametric simple regression to Spearman Rank correlation estimates (Figure 6 and lines 395-398).

Reviewer #2: The study tackled an interesting topic on gaze behavior of the runners in facing the natural obstacles in terms of pre-planning vs. online planning which is important to the respective field, the statistical analyses and results were sound and the manuscript was well-written.

---

## [Decision Letter · Decision Letter 1]

28 Apr 2020

PONE-D-20-07008R1

Gaze-Behaviors of Runners in a Natural, Urban Running Environment

PLOS ONE

Dear Mr. Cullen,

Thank you for submitting your manuscript to PLOS ONE. After careful consideration, we feel that it has merit but does not fully meet PLOS ONE’s publication criteria as it currently stands. Therefore, we invite you to submit a revised version of the manuscript that addresses the points raised during the review process.

We would appreciate receiving your revised manuscript by Jun 12 2020 11:59PM. To enhance the reproducibility of your results, we recommend that if applicable you deposit your laboratory protocols in protocols.io, where a protocol can be assigned its own identifier (DOI) such that it can be cited independently in the future. For instructions see: http://journals.plos.org/plosone/s/submission-guidelines#loc-laboratory-protocols

We look forward to receiving your revised manuscript.

Kind regards,

Nizam Uddin Ahamed, PhD

Academic Editor

PLOS ONE

Reviewers' comments:

Reviewer's Responses to Questions

**Comments to the Author**

1. If the authors have adequately addressed your comments raised in a previous round of review and you feel that this manuscript is now acceptable for publication, you may indicate that here to bypass the “Comments to the Author” section, enter your conflict of interest statement in the “Confidential to Editor” section, and submit your "Accept" recommendation.

Reviewer #1: All comments have been addressed

2. Is the manuscript technically sound, and do the data support the conclusions?

Reviewer #1: Yes

3. Has the statistical analysis been performed appropriately and rigorously? 

Reviewer #1: Yes

4. Have the authors made all data underlying the findings in their manuscript fully available?

Reviewer #1: Yes

5. Is the manuscript presented in an intelligible fashion and written in standard English?

Reviewer #1: Yes

6. Review Comments to the Author

Reviewer #1: The revision of the manuscript reasonably address all the comments with one exception. Regarding the switch from using Pearson correlation to Spearman rank correlation, corresponding changes/edits shall be made at the section of ‘Data analysis’ (lines 339-344) and the figure illustrating the correlation between fixation number and running speed shall be updated to reflect the tested (i.e., rank) instead of original variable metrics metrics.

I would endorse the acceptance of this manuscript after seeing these changes, which is necessary for a coherently presented manuscript.

7. PLOS authors have the option to publish the peer review history of their article (what does this mean?). If published, this will include your full peer review and any attached files.

Reviewer #1: Yes: Sicong Liu

---

## [Author Response · Author response to Decision Letter 1]

28 Apr 2020

We want to thank the editor, Dr. Liu, and our anonymous editor for their comments. We were able to accommodate your suggestions and we think that the changes have improved the manuscript. We are grateful for the opportunity to resubmit our manuscript for publication. 

Response to Reviewers

Reviewer #1: The revision of the manuscript reasonably address all the comments with one exception. Regarding the switch from using Pearson correlation to Spearman rank correlation, corresponding changes/edits shall be made at the section of ‘Data analysis’ (lines 339-344) and the figure illustrating the correlation between fixation number and running speed shall be updated to reflect the tested (i.e., rank) instead of original variable metrics.

I would endorse the acceptance of this manuscript after seeing these changes, which is necessary for a coherently presented manuscript.

Thank you for your comment, we are sorry we did not address this point fully in our last revision. We have adjusted line 339 to state “Spearman’s rank correlation” rather than “ordinary least squares regression.” We have also removed the regression line from the Spearman’s rank correlation in Figure 6.

---

## [Decision Letter · Decision Letter 2]

30 Apr 2020

Gaze-Behaviors of Runners in a Natural, Urban Running Environment

PONE-D-20-07008R2

Dear Dr. Cullen,

We are pleased to inform you that your manuscript has been judged scientifically suitable for publication and will be formally accepted for publication once it complies with all outstanding technical requirements.

With kind regards,

Nizam Uddin Ahamed, PhD

Academic Editor

PLOS ONE

Additional Editor Comments (optional):

Reviewers' comments:

Reviewer's Responses to Questions

**Comments to the Author**

1. If the authors have adequately addressed your comments raised in a previous round of review and you feel that this manuscript is now acceptable for publication, you may indicate that here to bypass the “Comments to the Author” section, enter your conflict of interest statement in the “Confidential to Editor” section, and submit your "Accept" recommendation.

Reviewer #1: All comments have been addressed

2. Is the manuscript technically sound, and do the data support the conclusions?

Reviewer #1: Yes

3. Has the statistical analysis been performed appropriately and rigorously? 

Reviewer #1: Yes

4. Have the authors made all data underlying the findings in their manuscript fully available?

Reviewer #1: Yes

5. Is the manuscript presented in an intelligible fashion and written in standard English?

Reviewer #1: Yes

6. Review Comments to the Author

Reviewer #1: (No Response)

7. PLOS authors have the option to publish the peer review history of their article (what does this mean?). If published, this will include your full peer review and any attached files.

Reviewer #1: Yes: Sicong Liu

---

## [Editor Report · Acceptance letter]

4 May 2020

PONE-D-20-07008R2 

Gaze-Behaviors of Runners in a Natural, Urban Running Environment 

Dear Dr. Cullen:

I am pleased to inform you that your manuscript has been deemed suitable for publication in PLOS ONE. Congratulations! Your manuscript is now with our production department. 

With kind regards,

on behalf of

Dr. Nizam Uddin Ahamed 

Academic Editor

PLOS ONE